# Prevalence of HIV drug resistance, its correlates and common mutations among people living with HIV failing on ART in northern Uganda: A cross-sectional study

**Francis Kayinda**[1,2]*, **Phyllis Awor**[1], **Twaha Mahaba**[2‡], **Alex M. Muganzi**[2‡], **Joanita Kigozi**[2‡], **Patrick Odong Olwedo**[3‡], **Esther M. Nasuuna**[2], **Robert Mutumba**[4‡], **Rhoda Wanyenze**[1]

**1** Makerere University School of Public Health, Kampala, Uganda, **2** Infectious Diseases Institute, Makerere University College of Health science, Kampala, Uganda, **3** Medical Department, Yumbe Regional Referral Hospital, Yumbe, Uganda, **4** STD/AIDS control program, Ministry of Health, Kampala, Uganda

☙ These authors contributed equally to this work.
‡ These authors also contributed equally to this work.
* kaysfrancis@gmail.com, fkayinda@idi.co.ug

## Abstract

### Background

HIV drug resistance (HIVDR) poses a challenge to managing people living with HIV (PLHIV), particularly among those experiencing virological failure (VF). The West-Nile region of Uganda faces HIV treatment challenges and has a high virological failure rate. We estimated the prevalence of HIV drug resistance, described the HIV drug resistance mutations and evaluated the factors associated with HIVDR among PLHIV with virological failure in the West-Nile region of Uganda.

### Methods

We conducted a retrospective cross-sectional analysis of HIVDR data in the West-Nile region of Uganda across the 161 health facilities that offer comprehensive Anti-retroviral therapy (ART) services. All PLHIV, regardless of age, who had been on ART for at least one-year, experienced virological failure and underwent an HIVDR test between 1st January 2021–30th December 2023 were included in the study. Demographic and clinical data were extracted from the National HIVDR database. HIVDR was defined as having at least one mutation with a penalty score of ≥15. PLHIV were characterized based on age, gender and clinical history. Logistic regression models determined factors associated with HIVDR with a p-value of <0.05 considered significant.

**Data availability statement:** The data used in this study is available on the Ugandan Ministry of Health HIV Drug Resistance Database but is not publicly accessible due to patient confidentiality policies. Access may be granted upon request to the Ugandan Ministry of Health through the Director General, Health Services. The email address for the data request is ps@health.go.ug and the website to the HIVDR data is https://hivdr.cphluganda.org/userAuth/login.

**Funding:** The author(s) received no specific funding for this work.

**Competing interests:** The authors have declared that no competing interests exist.

## Results

A total 295 records were analyzed. Of these, majority were female (56.6%) and adults aged ≥20 years (49.2%). The median age was 19 (Inter quartile range [IQR]: 13–41) years, and median duration on ART was 8 (IQR: 5–10) years. Overall, 218 (73.9%) had HIVDR with 66% of subjects having Non-nucleoside reverse transcriptase (NNRTI) mutations. M184V/I (50%), K103N (34%) and TAMS (26%) were the commonest mutations. Resistance to Etravirine (27%) was higher than that of Dolutegravir (12%) and Darunavir (5%). After accounting for gender, age, Nucleoside reverse transcriptase inhibitor (NRTI) anchor drug, ART regimen type and World Health Organization (WHO) clinical stage of the participants; long duration on ART (aOR=; 1.15 95%CI 1.05–1.26 p = 0.003), adolescents failing on first line (aOR=; 3.80 95%CI 1.02–14.08 p = 0.046) and participants failing on 2nd line (aOR=; 3.64 95%CI 1.18–11.21 p = 0.024) as indications for the HIVDR test, and the year of HIVDR sample collection (aOR=; 0.21 95%CI 0.07–0.69 p = 0.010), were independently associated with HIVDR mutations.

## Conclusion

The study found a high HIVDR prevalence strongly associated with long ART duration which is likely to lead to increased ART treatment failure rates. The high Etravirine resistance and increasing Dolutegravir resistance are likely to complicate future treatment options while low Darunavir resistance makes it a future third-line treatment option. Strengthening routine resistance surveillance, timely VL monitoring, and adherence support are critical to mitigating drug resistance and preserving ART effectiveness among PLHIV in the West-Nile region.

## Introduction

HIV drug resistance (HIVDR) is a growing challenge to the long-term success of antiretroviral therapy (ART) and the overall goal of HIV epidemic control. HIVDR arises when genetic mutations in HIV reduce the effectiveness of ART, resulting in virological failure (VF), increased transmission of resistant strains, and limited future treatment options. HIVDR can be transmitted (TDR), occurring before ART initiation, or acquired during treatment [1,2]. If unaddressed, HIVDR could lead to millions of deaths, new difficult-to-treat variants, and increased healthcare costs due to switching to more toxic and expensive regimens [3].

HIV remains a major global epidemic with 39.9 million people living with HIV (PLHIV) by the end of 2023, including 1.3 million new infections and over 630,000 deaths [4]. Sub-Saharan Africa (SSA) bears the largest burden, with 25.9 million PLHIV [4,5]. Uganda had an estimated 1.5 million PLHIV in 2023 with 38,000 new HIV infections and 20,000 deaths [6]. ART has significantly improved health outcomes for PLHIV who achieve HIV viral suppression through

maintaining immune function which leads to decreased mortality and morbidity from opportunistic infections. It also helps achieve epidemic control through the principle of undetectable = untransmissible (U = U) [7]. Uganda has made substantial progress in ART expansion, achieving a treatment coverage rate of 84% [7]. However, the national viral suppression rate remains suboptimal, estimated at 79% overall, 75% among adult males, and 63% among children and adolescents below 15 years, falling short of the Joint United Nations Program for AIDS (UNAIDS) 95-95-95 target [4,8].

Despite these achievements, the growing prevalence of HIVDR poses a serious threat to ART effectiveness. A World Health Organization (WHO) study (2004–2021) in 66 countries, including Uganda, found an 83.1% prevalence of acquired resistance among adults on first-line ART with virological failure (>1,000 copies/ml) [9]. In Uganda, a 2018 national survey reported 90.4% HIVDR among non-suppressed PLHIV [10], with other studies reporting rates ranging from 73.2% to 84.6% [11–13]. WHO (2014–2020) also reported the highest global TDR in Non-nucleoside Reverse Transcriptase Inhibitors (NNRTIs) (12.9%), followed by Nucleoside Reverse Transcriptase Inhibitors (NRTIs) (5.4%), Integrase strand transfer Inhibitors (INSTIs) (0.6%), and Protease Inhibitors (PIs) at (0.4%) [9]. A similar pattern was reported in Africa with NNRTIs at 15.4%, NRTIs at 6.1%, PIs at 0.3% and INSTIs at 0.1% [9]. In Uganda, NNRTI resistance was 15.4%, NRTIs at 5.1%, and PIs at 1%, while INSTIs remained largely unassessed due to low uptake. Other Ugandan studies reported TDR rates ranging from 5.9% to 18.2% [9].

The West Nile region of Uganda has a low HIV viral suppression rate (77%), compared to the national average [8]. The region faces unique challenges including a significant refugee population (22%), high levels of migration, bordering conflict-affected countries (South Sudan and Democratic republic of Congo), and high levels of poverty. These factors create social, economic, and health vulnerabilities which may impact ART adherence and the emergence of HIVDR [14–16]. Despite these concerns, there is limited data on the prevalence, specific mutations, and associated factors for HIVDR in this region. This study estimated the prevalence of HIVDR, described common mutations, and evaluated factors associated with HIVDR among non-suppressed PLHIV in the West Nile region of Uganda.

## Materials and methods

### Study design and participants

We conducted a retrospective cross-sectional study in the West-Nile region of Uganda across the 161 health facilities that offer comprehensive ART services [17]. The region had a population of 3.6 million people in 2023 and an estimated 49,000 PLHIV [15]. Study participants included all PLHIV, regardless of age, who had been on ART for at least one year, had a non-suppressed HIV viral load (>1000 copies/ml) and underwent an HIVDR test between 1st January 2021–30th December 2023.

### HIV drug resistance testing in Uganda

All PLHIV on ART in Uganda receive a VL test annually. Those with a non-suppressed VL undergo intensive adherence counselling (IAC) for three months with a repeat VL at the end. Individuals with a persistent non- suppressed VL are eligible for an HIVDR test. A Plasma or dried blood spot (DBS) sample is collected and referred to the HIVDR testing facility at a central national laboratory. Here, RNA extraction is done using commercial kits followed by reverse transcription and Polymerase Chain Reaction (PCR) amplification. PCR is a molecular technique used to amplify specific DNA sequences by generating millions of copies from a small initial sample. This step is crucial for increasing the quantity of genetic material required for sequencing. Sequencing, where individual mutations are identified using either Sanger sequencing or Next generation sequencing is then carried out. Mutations are identified and interpreted using the Stanford HIV drug resistance database [18–20]. The results of the HIVDR test are uploaded onto the national database together with the patients clinical and demographic information.

## Operational definitions

Penalty score: A penalty score is the value assigned to each HIV mutation to estimate the level of resistance it brings to a particular drug. Penalty scores of 0–9, represent no resistance; 10–14 represent potential low-level resistance; 15–29 represent low level resistance; 30–59 represent intermediate resistance and scores of ≥ 60 represent high level resistance [20]. The penalty score of ≥ 15 was used in this study to determine which mutations were drug resistant.

HIV Drug resistance: This was defined as the presence of a penalty score ≥15 assigned to an ARV drug using the Stanford HIV dB algorithm [20,21]. A participant with atleast one HIV drug resistant mutation to any ART drug was categorized as having HIVDR.

Virological failure: Virological failure was defined as two consecutive viral load results ≥ 1000 copies/ml despite undergoing at least three consecutive IAC sessions with good adherence between the tests [22].

VL Non suppression: This was defined as viral load copies ≥ 1000 copies per ml [22].

Age groups: Because of the varying risks of HIV acquisition, transmission, and disease progression, participants' age was classified as follows: Children (<10 years), Adolescents (10–19 years) and Adults (≥20 years) [7,23,24].

## Data collection

Census sampling was used to include all PLHIV with HIV DR results. The data abstraction tool was pretested on 30 individual participants (10% of the participants' data) before it was used to collect the demographic and clinical data. Data were accessed from the national database on 16th May 2024. Demographic data included age and sex. Clinical data included viral load copies preceding the DR test, WHO clinical stage, ART regimen class at the time of sample collection, NRTI anchor drug of the regimen, the full ART regimen, duration on ART, the line of ART treatment, indication for the HIVDR test, year of HIVDR test and actual mutations per participant.

## Data analysis

We performed statistical analysis using Stata SE version 14.2, developed by Stata Corp LLC in the United States. To assess the distributions of the independent variables (clinical and demographic), we used the Shapiro-Wilk test to assess the normality of continuous variables. Continuous variables were summarized using medians and interquartile ranges while categorical variables were presented as frequencies and proportions. We performed a Chi-square test, a Wilcoxon rank-sum test and bivariable logistic regression analysis to assess association of each of the independent potential correlates with the outcome. Variables that had an odds ratio (OR) with $P \leq 0.2$ were included in multivariable logistic regression model together with age and sex as potential confounders. Results were presented as adjusted odds ratios (aORs) with 95% confidence interval (CI). Variables were considered significantly associated with the outcome if the P value for the aOR was less than or equal to 0.05.

## Ethics approval and consent to participate

This study was conducted in accordance with ethical principles of the Helsinki declaration. Institutional ethical approval was obtained from the Makerere University School of Public Health Research and Ethics Committee with reference number 366. Permission to use national data was granted by the Ugandan Ministry of Health through the Director General, Health Services. The study used de-identified secondary data, and the authors did not have access to information that could identify individual participants during or after data collection, hence the need for individual participant consent was waived by the ethics committee. All filled forms were kept safely in a waterproof locked cabinet where they could only be accessed by the principal investigator.

## Results

During the study period, 363 PLHIV had virological failure but only 295 (81.3%) had HIV DR results and were considered for the study (Fig 1).

### Baseline participant characteristics and HIVDR

Of the 295 participants, 167 (56.6%) were female and the median age was 19 years (Inter quartile range (IQR: 13–41). The participants had been on ART for a median 8 years (IQR: 5–10) and 59.3% were on INSTIs at sample collection. Majority of participants (65.4%) underwent HIVDR testing in 2023 following revised national criteria to test all PLHIV failing on ART regardless of treatment line (Table 1).

### HIVDR mutations among participants

HIVDR mutations were present in 218 participants (73.9%: 95% Confidence Interval (CI) 68.9–78.9%). Majority of the participants (66%) had NNRTI mutations while 55% had NRTI mutations (Table 2).

### HIV drug resistance across drug classes

HIVDR mutations were observed across different drug classes in the same individuals where 55% (161) of all participants had resistance to at least two drug classes and 1% (03) of the participants had resistance across all the 4 drug classes (Table 3).

### HIVDR among children and adolescents

Among the 295 participants, 150 (50.8%) were children and adolescents living with HIV (CALHIV) under 20 years of age. Of these, 37 (12.5%) were children under 10 years, and 113 (38.3%) were adolescents (10–19 years). Overall, 76% of CALHIV had HIVDR, with children having a higher prevalence (81.1%) compared to adolescents (74.3%). NNRTI resistance was the most common, with resistant mutations detected among 70% of CALHIV while resistant mutations to INSTIs and PIs were present among 12.0% and 11.3% in CALHIV respectively. Resistance to etravirine was the highest

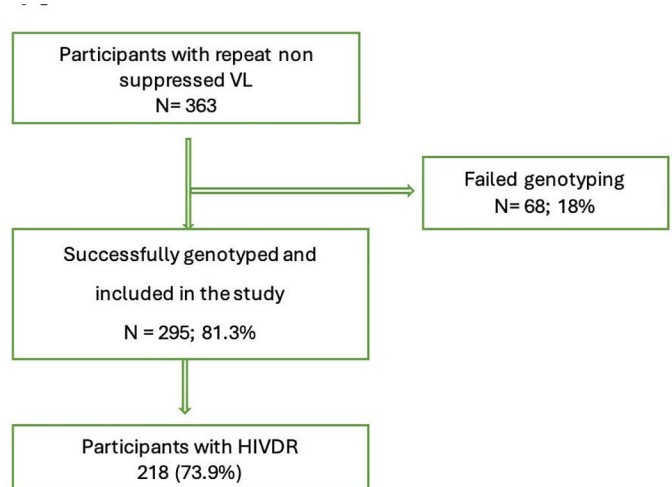

**Fig 1. A flow chart that shows the participants who were included in the study.**

**Table 1. Baseline Participant characteristics and HIVDR.**

| Characteristic | Frequency % (n/N) N = 295 | Number with HIVDR mutations (%) | Number with no HIVDR mutations (%) | P-Value (Chi-square Test) |
|---|---|---|---|---|
| **Gender** | | | | |
| Female | 167 (56.6) | 124 (74.3) | 43 (25.7) | 0.875 |
| Male | 128 (43.4) | 94 (73.4) | 34 (25.6) | |
| **Age Group** | | | | |
| Children (<10 years) | 37 (12.5) | 30 (81.1) | 07 (18.9) | 0.508 |
| Adolescents (10–19 years) | 113 (38.3) | 84 (74.3) | 29 (25.7) | |
| Adults (20+years) | 145 (49.2) | 104 (71.7) | 41 (28.3) | |
| **NRTI Anchor at time of sample collection** | | | | |
| ABC/3TC | 87 (29.5) | 67 (77.0) | 20 (23.0) | **0.041*** |
| AZT/3TC | 78 (26.4) | 64 (82.1) | 14 (17.9) | |
| TDF/3TC | 130 (44.1) | 87 (66.9) | 43 (33.1) | |
| **Type of regimen at time of sample collection** | | | | |
| NNRTI | 17 (5.7) | 14 (82.3) | 03 (17.7) | **0.009*** |
| PI | 103 (34.9) | 86 (83.5) | 17 (15.5) | |
| INSTI | 175 (59.3) | 118 (67.4) | 57 (32.6) | |
| **ART regimen at time of sample collection** | | | | |
| TDF/3TC/DTG | 103 (34.9) | 63 (61.2) | 40 (38.8) | **0.013*** |
| AZT/3TC/DTG | 35 (11.9) | 29 (82.9) | 06 (17.1) | |
| ABC/3TC/DTG | 37 (12.5) | 26 (70.3) | 11 (29.7) | |
| ABC/3TC/LPVr | 45 (15.3) | 37 (82.2) | 08 (17.8) | |
| TDF/3TC/ATVr | 12 (4.1) | 10 (83.3) | 02 (16.7) | |
| AZT/3TC/ATVr | 21 (7.1) | 17 (81.0) | 04 (19.0) | |
| OTHERS | 42 (14.2) | 36 (85.7) | 06 (14.3) | |
| **VL Leading to the HIVDR test (copies/ml)** | | | | |
| 1,000-9,999 | 92 (31.2) | 63 (68.5) | 29 (31.5) | 0.122 |
| 10,000-99,999 | 128 (43.4) | 102 (79.7) | 26 (20.3) | |
| 100,000-999,000 | 66 (22.4) | 45 (68.2) | 21 (31.8) | |
| 1,000,000 + | 09 (3.1) | 08 (88.9) | 01 (11.1) | |
| **Treatment Line** | | | | |
| First line | 145 (49.2) | 94 (64.8) | 51 (35.2) | **0.002*** |
| Second line | 138 (46.8) | 113 (81.9) | 25 (18.1) | |
| Third line | 12 (4.0) | 11 (91.7) | 01 (8.3) | |
| **HIVDR test Indication** | | | | |
| Children failing on 1st line | 34 (11.5) | 27 (79.4) | 07 (20.6) | **0.001*** |
| Adolescent failing on 1st line | 44 (14.9) | 32 (72.7) | 12 (27.3) | |
| Adult failing on 1st Line | 67 (22.7) | 35 (52.2) | 32 (47.8) | |
| Patients failing on 2nd Line | 138 (47.8) | 113 (81.9) | 25 (18.1) | |
| Patient failing on 3rd line | 12 (4.1) | 11 (91.7) | 01 (8.3) | |
| **WHO clinical stage** | | | | |
| I/II | 242 (82.0) | 175 (72.3) | 67 (27.7) | 0.228 |
| III/IV | 53 (18.0) | 43 (81.1) | 10 (18.9) | |
| **Year of Sample collection** | | | | |
| 2021 | 51 (17.3) | 46 (90.2) | 05 (9.8) | **0.001*** |
| 2022 | 51 (17.3) | 43 (84.3) | 08 (15.7) | |
| 2023 | 193 (65.4) | 129 (66.8) | 64 (33.2) | |

*(Continued)*

**Table 1.** (Continued)

| Characteristic | Frequency % (n/N) N=295 | Number with HIVDR mutations (%) | Number with no HIVDR mutations (%) | P-Value (Chi-square Test) |
|---|---|---|---|---|
| **Continuous variables** | | | | |
| Characteristic | **Median (IQR) (N=295)** | **Median (IQR) HIVDR Present (n=218)** | **Median (IQR) HIVDR Absent (n=77)** | **P-value (Wilcoxon rank-sum test)** |
| Age (years) | 19 (13–41) | 19 (13–41) | 22 (13–37) | 0.442 |
| Total duration on ART (years) | 8 (56789–10) | 8 (6789–10) | 7 (5678–9) | **0.001*** |
| VL leading to HIVDR test (copies/mL) | 25,300 (7,160–106, 000) | 25,000 (7,790–98,500) | 27,300 (4,890–108, 000) | 0.819 |

IQR, Interquartile range; ABC, Abacavir, 3TC, Lamivudine; AZT, Zidovudine; TDF, Tenofovir; DTG, Dolutegravir; ATVr, Atazanavir ritonavir; LPVr, Lopinavir ritonavir; NNRTI, Non-nucleoside reverse transcriptase inhibitor; NRTI, nucleo(t)side reverse transcriptase inhibitor; PI, Protease Inhibitor; INSTI, Integrase strand transfer inhibitor; WHO, World Health Organization; VL, Viral Load; ART, Anti-retroviral therapy, HIVDR, HIV drug resistance.

* Statistically significant variables with p ≤ 0.05.

**Table 2. HIVDR Mutations according to drug classes.**

| Regimen | Number and Proportion of HIVDR Mutations among all 295 participants. |
|---|---|
| NNRTIs | 196 (66%) |
| NRTIs | 161 (55%) |
| PIs | 46 (16%) |
| INSTIs | 40 (14%) |
| **Total** | **218 (73.9%)** |

NNRTI, Non-nucleoside reverse transcriptase inhibitor; NRTI, nucleo(t)side reverse transcriptase inhibitor; PI, Protease Inhibitor; INSTI, Integrase strand transfer inhibitor.

**Table 3. Patterns of resistance across drug classes.**

| Drug combinations | Frequency of HIVDR | Proportion among all 295 participants |
|---|---|---|
| One drug class | 57 | 19% |
| Two drug classes | 102 | 55% |
| Three drug classes | 56 | 19% |
| Four drug classes | 03 | 01% |

HIVDR, HIV drug resistance.

among third-line candidate drugs, detected in 28% of CALHIV followed by resistance to darunavir (9.3%) and dolutegravir (6.0%) (Table 4).

## HIVDR mutations among participants

The M184V was the commonest mutation observed in NRTI class appearing among 146 (50%) participants. The thymidine analogue mutations (TAMs) were observed among 77 (26%) participants and Type 2 pattern TAMs were more common (56% of all TAMs) than type 1 pattern TAMs. K65R was only observed among only 4 (1%) participants. Among NNRTIs, K103N/S mutations were the commonest appearing among 99 (34%) participants. Overall, 81 (27%)

**Table 4. HIVDR among children and Adolescents.**

| Characteristic | Children (<10 yrs) Frequency (%) | Adolescents (10–19 yrs) Frequency (%) | Total CALHIV Frequency (%) |
|---|---|---|---|
| Number of Participants | 37 (12.5) | 113 (38.3) | 150 (50.8) |
| Presence of HIVDR mutations | | | |
| HIVDR present | 30 (81.1) | 84 (74.3) | 114 (76.0) |
| NNRTI Mutations | 28 (75.7) | 77 (68.1) | 105 (70.0) |
| NRTI Mutations | 22 (59.5) | 65 (57.5) | 87 (58.0) |
| INSTI Mutations | 04 (12.9) | 14 (12.4) | 18 (12.0) |
| PI Mutations | 04 (12.9) | 13 (11.5) | 17 (11.3) |
| Patterns of resistance across drug classes | | | |
| One drug class | 08 (21.6) | 19 (16.8) | 27 (24.7) |
| Two drug classes | 16 (43.2) | 50 (44.2) | 66 (44.0) |
| Three drug classes | 06 (16.2) | 14 (12.4) | 20 (13.3) |
| Four drug classes | 00 (00%) | 02 (1.7) | 02 (1.3) |
| Resistance to potential third-line drugs | | | |
| Etravirine Resistance | 10 (27.0) | 32 (28.3) | 42 (28.0) |
| Dolutegravir Resistance | 03 (8.1) | 06 (5.3) | 09 (6.0) |
| Darunavir Resistance | 02 (5.4) | 12 (10.6) | 14 (9.3) |

HIVDR, HIV drug resistance; NNRTI, Non-nucleoside reverse transcriptase inhibitor; NRTI, nucleo(t)side reverse transcriptase inhibitor; PI, Protease Inhibitor; INSTI, Integrase strand transfer inhibitor.

participants had resistance to Etravirine. The darunavir sparing M46V/A mutations were the commonest among Protease Inhibitors (PIs) appearing in 20 (7%). Overall, 14 (5%) participants had resistance to darunavir. The E138K/A and G118R mutations were the commonest amongINSTIs occurring in 18 (6%) and 12 (4%) participants respectively and in total 35 (12%) participants had resistance to Dolutegravir (DTG). 24 (8%) participants had a mixture of wild type virus with the mutated virus (Fig 2).

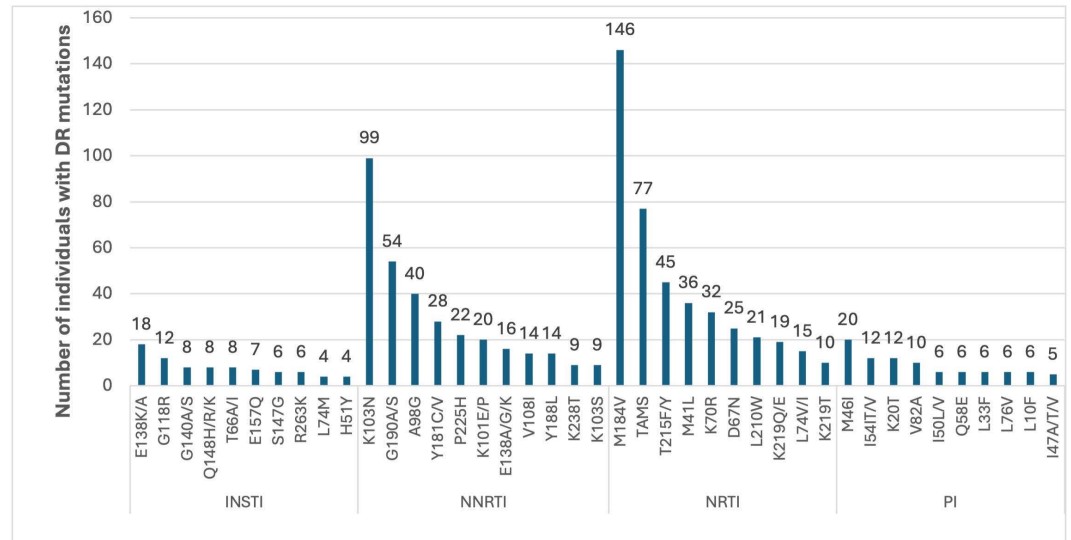

**Fig 2. Shows the commonest HIVDR mutations among the 218 participants with atleast one mutation categorized by drug class.**

## Factors associated with HIVDR

After accounting for gender, age, NRTI anchor drug, ART regimen type and WHO clinical stage of the participants; long duration on ART, indication for the HIVDR test and the year of HIVDR sample collection were independently associated with HIVDR mutations. The odds of HIVDR significantly increased by 15% for each additional year an individual spent on ART (aOR=; 1.15 95%CI 1.05–1.26 p = 0.003). For the indication for the HIVDR test, adolescents failing on first line (aOR=; 3.80 95%CI 1.02–14.08 p = 0.046) and participants failing on 2nd line (aOR=; 3.64 95%CI 1.18–11.21 p = 0.024) were found to be significantly associated with HIVDR. The HIVDR tests conducted in 2023 (aOR=; 0.23 95%CI 0.07–0.76 p = 0.01) were also significantly associated with HIVDR (Table 5).

## Discussion

We set out to estimate the prevalence of HIVDR, describe common mutations, and associated factors among non-suppressed PLHIV in the West Nile region of Uganda. We found that 73.9% had at least one mutation. The commonest mutations were the NNRTI mutations mostly K103N/S and G190A/S/E. The associated factors were indication for the HIVDR test, duration on ART and the year of HIVDR sample collection.

The observed prevalence (73.9%) is lower than the global and African estimates of 83.5% as reported by WHO in eight countries that included adult PLHIV on ART for four or more years [9]. It is also below what was found in a Ugandan study among 1,064 adult participants on first line for ≥ 48 months that found a prevalence of 90.4% [10]. Estimates from neighboring countries of Rwanda and Kenya, were 92.7% and 82.3% respectively, as reported in studies that included 500 PLHIV that were failing on first line [25,26]. We postulate that the lower prevalence we found can be explained by the suboptimal adherence associated with intermittent drug use within the region which could lead to viral failure without sufficient selective pressure to drive resistance mutations. This means continued nonadherence will lead to mutations among the PLHIV and improved adherence is likely to prevent development of mutations [2].

The commonest NNRTI mutations (K103N/S and G190A/S/E) found in 66% of the participants confer resistance to efavirenz and nevirapine with a single mutation conferring resistance to all first-generation drugs [27–29]. These two drugs were part of the recommended first line drugs in Uganda until 2018 [22]. The high rates of NNRTI resistance is consistent with other studies conducted in similar settings such as Uganda (84.6%), Rwanda (90.4%), Kenya (81%) among PLHIV failing on treatment [9,10,25,26]. The high rate of etravirine resistance (27%) conferred by mutations such as G190A/S and Y181C is likely to limit third line treatment options [23]. The high rates of M184V/I mutations (50%) from NRTIs is attributed to continued usage of lamivudine as a backbone in all regimens in SSA since the presence of this mutation reduces the viral replicative fitness and increases susceptibility to Tenofovir and Zidovudine (AZT) [30]. The prolonged use of AZT included currently as part of the recommended second line of treatment in Uganda explains the high prevalence of TAMs (26%) among participants [23]. The lower mutation rates in PIs (16%) and INSTI (14%) is due the higher genetic barrier to resistance of boosted PI-based ART and INSTIs [31] and the fact that INSTIs have just recently been introduced into the HIV program as first line drugs and participants have been exposed to them for a shorter time [22]. Other studies reported similar low prevalences: a review of 20 studies in 11 SSA countries (PI resistance 17%) [32] and a review of 88 publications worldwide (INSTI resistance 1.5%) among ART-experienced PLHIV [33]. The mutations mainly spared darunavir (5%) the most commonly used drug in third line of treatment, however, the dolutegravir resistance of 12% is high for a drug with a high genetic barrier to resistance and that has only been recently scaled up [22]. It is an issue of concern to be noted by the HIV program since it is likely to limit HIV treatment options [23]. This increasing resistance could be due to unintended DTG dual or monotherapy due to failure of other low genetic barrier NRTIs within the treatment regimen which greatly increases the risk of resistance [34,35]. The association between duration on ART and HIVDR is supported by the notion that prolonged ART exposure increases the risk of emergence of acquired drug resistant mutations [36] and could also be a reflection of the time an individual has spent on a suboptimal regimen or with adherence challenges which are associated with insufficient drug pressure in the body which leads to HIVDR [37]. A systematic review of seven studies conducted in Africa, Asia and south America found a similar association

**Table 5. Correlation of baseline participant characteristics with odds of HIVDR mutations.**

| Characteristic | Unadjusted Odds Ratio (95% CI) | *P*-value | Adjusted Odds Ratio (95% CI) | *P-Value* |
|---|---|---|---|---|
| **Duration on ART** | | | | |
| Median (8 years) | 1.15 (1.06-1.23) | 0.001 | *1.15 (1.05-1.26)* | ***0.003**** |
| **Age** | | | | |
| Children (<10 years) | Referent | | | |
| Adolescents (10–19 years) | 0.68 (0.27- 1.70) | 0.406 | *0.47 (0.13-1.70)* | *0.247* |
| Adults (20 + years) | 0.59 (0.24 - 1.45) | 0.253 | *1.26 (0.28-5.68)* | *0.764* |
| **Gender** | | | | |
| Female | 1.04 (0.62-1.76) | 0.875 | *1.26 (0.69-2.29)* | *0.456* |
| Male | Referent | | | |
| **NRTI Anchor at time of sample collection** | | | | |
| ABC/3TC | 1.66 (0.89-3.07) | 0.110 | *1.37 (0.54-3.49)* | *0.536* |
| AZT/3TC | 2.26 (1.14-4.48) | 0.02 | *1.24 (0.48-3.17)* | *0.661* |
| TDF/3TC | Referent | | | |
| **Type of regimen at time of sample collection** | | | | |
| NNRTI | 2.25 (0.62-8.16) | 0.216 | *1.11 (0.25-4.93)* | *0.894* |
| PI | 2.44 (1.32-4.49) | 0.004 | *0.66 (0.26-1.67)* | *0.382* |
| INSTI | Referent | | | |
| **ART regimen at time of sample collection** | | | | |
| TDF/3TC/DTG | Referent | | | |
| AZT/3TC/DTG | 3.07 (1.17-8.04) | 0.023 | | |
| ABC/3TC/DTG | 1.50 (0.67-3.37) | 0.325 | | |
| ABC/3TC/LPVr | 2.94 (1.24-6.95) | 0.014 | | |
| TDF/3TC/ATVr | 3.17 (0.66-15.24) | 0.149 | | |
| AZT/3TC/ATVr | 2.70 (0.85-8.60) | 0.093 | | |
| OTHERS | 3.81 (1.47-9.86) | 0.006 | | |
| **VL Leading to the HIVDR test (copies/ml)** | | | | |
| 1,000-9,999 | Referent | | | |
| 10,000-99,999 | 1.81 (0.98-3.34) | 0.060 | | |
| 100,000-999,000 | 0.99 (0.50-1.95) | 0.968 | | |
| 1,000,000 + | 3.68 (0.44-30.83) | 0.229 | | |
| **Treatment Line** | | | | |
| First line | Referent | | | |
| Second line | 2.45 (1.41-4.26) | 0.001 | | |
| Third line | 5.97 (0.75-47.54) | 0.092 | | |
| **HIVDR Indication** | | | | |
| Children failing on 1st line | 3.52 (1.35-9.21) | 0.01 | *4.36 (0.84 - 22.67)* | *0.080* |
| Adolescent failing on 1st line | 2.44 (1.08-5.53) | 0.033 | *3.80 (1.02-14.08)* | ***0.046**** |
| Adult failing on 1st Line | Referent | | | |
| Patients failing on 2nd Line | 4.13 (2.17-7.89) | 0.000 | *3.64 (1.18-11.21)* | ***0.024**** |
| Patient failing on 3rd line | 10.06 (1.23-82.33) | 0.031 | *9.70 (0.90- 104.95)* | *0.061* |
| **WHO clinical stage** | | | | |
| I/II | Referent | | | |
| III/IV | 1.64 (0.78-3.46) | 0.189 | *2.00 (0.86-4.65)* | *0.107* |
| **Year of Sample collection** | | | | |
| 2021 | Referent | | | |
| 2022 | 0.58 (0.18 - 1.92) | 0.377 | *0.52 (0.15–1.81)* | *0.307* |

*(Continued)*

**Table 5.** (Continued)

| Characteristic | Unadjusted Odds Ratio (95% CI) | P-value | Adjusted Odds Ratio (95% CI) | P-Value |
|---|---|---|---|---|
| 2023 | 0.29 (0.08 - 0.58) | 0.002 | 0.23 (0.07 - 0.76) | **0.010*** |

OR, Odds ratio; CI, Confidence interval; ABC, Abacavir, 3TC, Lamivudine; AZT, Zidovudine; TDF, Tenofovir; DTG, Dolutegravir; ATVr, Atazanavir ritonavir; NNRTI, Non-nucleoside reverse transcriptase inhibitor; NRTI, nucleo(t)side reverse transcriptase inhibitor; PI, Protease Inhibitor; INSTI, Integrase strand transfer inhibitor.

* Statistically significant variables with p ≤ 0.05.

[38] while other similar studies in Uganda and Rwanda didn't find any association [12,26]. In case of the HIVDR test indication, factors such as peer pressure, stigma and discrimination, drug and alcohol abuse and non-disclosure stand out among adolescents leading to poor adherence [23] and hence development of HIVDR while participants on second-line of treatment have spent a longer time on treatment and have also undergone prior regimen changes due to probable or confirmed HIVDR [36]. Addressing adherence challenges among adolescents mainly through peer support and adolescent friendly services will be key in prevention of HIVDR [38]. Prior to 2023, the eligibility criteria for a HIVDR test in Uganda included only PLHIV failing on either second or third line of treatment, however, in 2023 the criteria was changed to include all PLHIV with treatment failure irrespective of the treatment line. This meant that tests conducted in 2023 following the change in eligibility criteria were less likely to have developed HIVDR since they had been on a failing regimen for a short duration [38]. The national viral suppression rates among children were estimated to be as low as 63% compared to the national average of 79% underscoring the heightened vulnerability within the HIV treatment cascade. In this study, age was not statistically associated with HIVDR despite children under 10 years showing the higher proportion of HIVDR at 81.1% compared to other age groups (Table 1). This apparent discrepancy may be explained by suboptimal ART adherence that is more pronounced among children leading to viral failure without exerting sufficient selective pressure to drive resistance mutations in short term, although persistent poor adherence may eventually result in the development of HIVDR [4,8].

The study has a larger sample size of participants compared to previous studies conducted within Uganda and has resistance data against INSTIs which were not present in the country. Our study has several limitations. Firstly, the HIVDR data prior to current ART initiation was not available so we could not rule out the possibility of transmitted drug resistance. Secondly we were limited to the data available from medical records and patient histories. This means that not all factors potentially associated with HIV drug resistance, were assessed. Furthermore, as our study is a retrospective study using secondary data, it is subject to biases related to omissions and commissions during initial data collection and entry into the HIVDR database and these may impact the accuracy and reliability of our study.

## Conclusion

The study found a high prevalence of drug-resistant mutations among PLHIV failing on ART, strongly associated with long ART duration although INSTI and PI drug classes were mainly spared. This resistance is likely to lead to increased ART treatment failure rates and limited regimen options. The high Etravirine resistance and increasing Dolutegravir resistance are likely to complicate future treatment options while low Darunavir resistance makes it a future third-line treatment option. Strengthening routine resistance surveillance, timely VL monitoring, and adherence support are critical to mitigating drug resistance and preserving ART effectiveness. Additional research assessing the prevalence of transmitted drug resistance, or association between other factors like socio-cultural, viral and structural and HIVDR are recommended.

## Acknowledgments

The authors extend their gratitude to the Ugandan Ministry of Health, the Makerere University School of Public Health, and the Infectious Diseases Institute for supporting this research. Special thanks to the Central Public Health Laboratories for their role in HIV drug resistance testing.

## Author contributions

**Conceptualization:** Kayinda Francis.

**Data curation:** Kayinda Francis, Esther M. Nasuuna, Twaha Mahaba, Joanita Kigozi, Alex M. Muganzi, Rhoda Wanyenze.

**Formal analysis:** Kayinda Francis.

**Investigation:** Kayinda Francis.

**Methodology:** Kayinda Francis, Phyllis Awor, Rhoda Wanyenze.

**Project administration:** Robert Mutumba.

**Resources:** Patrick Odong Olwedo.

**Supervision:** Esther M. Nasuuna, Patrick Odong Olwedo, Phyllis Awor, Rhoda Wanyenze.

**Validation:** Esther M. Nasuuna, Joanita Kigozi, Phyllis Awor, Rhoda Wanyenze.

**Writing – original draft:** Kayinda Francis, Phyllis Awor, Rhoda Wanyenze.

**Writing – review & editing:** Kayinda Francis, Esther M. Nasuuna, Twaha Mahaba, Joanita Kigozi, Alex M. Muganzi, Patrick Odong Olwedo, Robert Mutumba, Phyllis Awor, Rhoda Wanyenze.

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
