## [Decision Letter · Decision Letter 0]

24 Mar 2025

Dear Dr. Francis,

Thank you for submitting your manuscript to PLOS ONE. After careful consideration, we feel that it has merit but does not fully meet PLOS ONE’s publication criteria as it currently stands. Therefore, we invite you to submit a revised version of the manuscript that addresses the points raised during the review process.

We look forward to receiving your revised manuscript.

Kind regards,

Felix Bongomin, MB ChB, MSc, MMed, FECMM

Academic Editor

PLOS ONE

Journal Requirements:

2**.** In the online submission form, you indicated that your data is available only on request from a third party. Please note that your Data Availability Statement is currently missing contact details for the third party, such as an email address or a link to where data requests can be made. Please update your statement with the missing information.

Reviewers' comments:

Reviewer's Responses to Questions

**Comments to the Author**

1. Is the manuscript technically sound, and do the data support the conclusions?

Reviewer #1: Yes

Reviewer #2: Partly

2. Has the statistical analysis been performed appropriately and rigorously?

Reviewer #1: No

Reviewer #2: Yes

3. Have the authors made all data underlying the findings in their manuscript fully available?

Reviewer #1: No

Reviewer #2: No

4. Is the manuscript presented in an intelligible fashion and written in standard English?

Reviewer #1: No

Reviewer #2: Yes

Reviewer #1: The authors carried out a retrospective cross-sectional study to assess the prevalence of HIV drug resistance, its associated factors, and common mutations among individuals living with HIV in Northern Uganda. To enhance the scientific rigor of the manuscript, please find my comments and suggestions below:

Abstract

Background: Last sentence, please specify the place as opposed to saying “in the region”

Methods: Similar comment for the methods section. In the methods section, it will be good to specify the population (everyone taking ART irrespective of age or what?). additionally, did you include everyone irrespective of ART duration? Data is a plural word and it should be followed by “were” as opposed to was. Please specify. How was HIVDR defined? It would begin this section by saying “We conducted a retrospective cross-sectional……………”. How did you characterize PLHIV? Be specific.

Results: Is HIVDR testing targeted in your region? If yes, then the prevalence of 73.9% might not be surprising. What other variables did you account for in your adjusted analysis? Please include them. You can say “After accounting for ……………, long duration on ART was significantly associated with presence of HIVDR mutations (aOR=; 1.15 95%CI 1.05 -1.26 p=0.003).”

Conclusion: replace “region” with a specific place being referred to. In the conclusion, there is no need of comparing with other previous findings. In this section, generally, you have to highlight your main findings and, the implication/s of those findings and end by giving recommendation/s. please revise this section accordingly.

Introduction

- In this section, I would first introduce the topic to the readers; in this case, HIVDR, give global and regional statistics.

- Replace amongst with among

Materials and Methods

- Say “retrospective cross-sectional…..”

- Include the age of the participants, e.g., PLHIV aged � 15 years, etc.

- Define PCR

- Data collection: How did you pre-test the data abstraction tool? Say data were (see my comment above)

- Ethics approval and consent to participate should be the last part of your materials and methods section. Please revise.

Data Analysis: state the test of normality used. Delete this part “The dependent variable was presence of at least one HIVDR mutation” and have a section in the materials and methods section where you will have “Operation definition” for your dependent/outcome/response variable.

Results

- “They had been on ART for a median 8 years”, who?

- Define in the methods section, preferably under “Operational definition” how age was categorized and cite.

- Table 1, I would name it “Baseline Participant characteristics and HIVDR”. Then, narrate the results of HIVDR also so you can guide the readers. On variable ‘HIVDR Indication”, why didn’t you aggregate 2nd and 3rd lines based on children, adolescents, and adults just the way you did for 1st line? In the footnote of the Table, please define all the abbreviations.

- For Table 1, authors should conduct a chi-square test/Fisher’s exact test between HIVDR (yes or no) and other independent categorical variables and a Wilcoxon rank-sum test for two medians. Create 5 columns; column 1 Characteristics, column 2 Frequency, column 3 Presence of HIVDR, column 4 Absence of HIVDR, and column 5 p-value. We can now know whether the proportions and medians are statistically significant.

- Presence of HIVDR mutations: define CI. Show these results in the Table “HIVDR mutations were observed across different drug classes in the same individuals where 55% (161) of all participants had resistance to at least two drug classes; 20% (59) had to at least three drug classes and 1% (03) of the participants had resistance across all the 4 drug classes. The commonest cross resistance was observed between NNRTI and NRTI drug classes (67%).”

- HIVDR Mutations amongst participants: Replace the word “amongst with among” throughout the document.

- I don’t think Figures 1 and 2 should be supplementary. They have to be in the main document.

- Factors associated with HIVDR: replace the word spends with spent. In the abstract, you have to mention all the variables which were significant at adjusted analysis. For 3rd line, the CI is too wide; I hope your discussion accounted for this imprecision. Make sure all the p-values have three decimal places in Table 3. The footnote of all your Tables should define the abbreviations.

Discussion

- AZT including, say “included”

- noted by the program- say HIV program

- This does not read well “For the indication for the HIVDR test”, replace. Maybe “In case of the HIVDR test indication”

- development HIVDR, here “of” is missing.

- Delete respectively “prior regimen changes due to probable or confirmed HIVDR respectively”

- Being a retrospective study, did you collect all the factors associated with HIV drug resistance e.g. CD4? You have only mentioned one limitation. Please include all the limitations inherent in retrospective studies.

Conclusion

- Don’t think 73.9% is lower. Here, tell us the meaning (low or high) of the proportions you have found without comparing with previous studies.

Reviewer #2: The manuscript had described a relatively lower prevalence of HIVDR among PLHIV with non-suppressive ART than other studies. It's notable that longer duration of ART was associated with presence of HIVDR mutations. However, this study had some shortcomings and should be reversed properly, then later considered if it is suit to published in this journal

Major concerns:

The topic should be more clear to define the study population that the prevalence of HIVDR was not among all PLHIV but among those with non-suppressive ART.

The study had included children and adolescent which might have different prevalence of virological failure, HIVDR and related factors, and deserve to be described and discussed disparately.

Minor concerns:

There was not any rates of virological failure among different groups, which would also be related to non-adherence.

What was the prevalence of HIVDR among ART naive PLHIV in the regions.

**Do you want your identity to be public for this peer review?** For information about this choice, including consent withdrawal, please see our Privacy Policy

Reviewer #1: No

Reviewer #2: No

---

## [Author Response · Author response to Decision Letter 1]

22 Apr 2025

PONE-D-25-09503

Title: Prevalence of HIV drug resistance, its correlates and common mutations among people living with HIV failing on ART in northern Uganda. A cross-sectional study.

# Editors comments

This was all actioned

The study used secondary data from the National HIVDR database. Details about the process of HIVDR testing have been described in the manuscript on Page 5 under the subtitle “HIV drug resistance testing in Uganda”.

We have adjusted the following sections to meet the requirements

• References and Acknowledgement to Heading 1 (Page 20 line 3 and 7)

• Study design and Operational definitions to Heading 2 (Page 2 line 5 and Page 3 line 1)

We have adjusted the following sections to meet the requirements. (Page 1 line 4-15)

• We modified the Title to sentence case format.

• We indicated symbol legends on each author.

• We modified author affiliations to the required format

2. In the online submission form, you indicated that your data is available only on request from a third party. Please note that your Data Availability Statement is currently missing contact details for the third party, such as an email address or a link to where data requests can be made. Please update your statement with the missing information.

The email address for the Ministry of Health contact has been indicated along with the link to the database. The email address for the data request is ps@health.go.ug and the website to the HIVDR data is https://hivdr.cphluganda.org/userAuth/login

Captions have been included for the 2 figures ( page 8 line 20 and page 12 line 7-8).

Reviewer #1: The authors carried out a retrospective cross-sectional study to assess the prevalence of HIV drug resistance, its associated factors, and common mutations among individuals living with HIV in Northern Uganda. To enhance the scientific rigor of the manuscript, please find my comments and suggestions below:

Abstract

Background: Last sentence, please specify the place as opposed to saying “in the region”

The region has been changed to reflect the West Nile region of Uganda. (Page 1 Line 21-22)

Methods: Similar comment for the methods section. In the methods section, it will be good to specify the population (everyone taking ART irrespective of age or what?). additionally, did you include everyone irrespective of ART duration? Data is a plural word, and it should be followed by “were” as opposed to was. Please specify. How was HIVDR defined? It would begin this section by saying “We conducted a retrospective cross-sectional……………”. How did you characterize PLHIV? Be specific.

The population has been specified as “All PLHIV, regardless of age, who had been on ART for at least one year, had virological failure and underwent an HIVDR test between 1st January 2021 to 30th December 2023’’ (Page 1 line 25)

HIVDR was defined as having at least one mutation with a penalty score of ≥15. (Page 2, line 3-4)

Results: Is HIVDR testing targeted in your region? If yes, then the prevalence of 73.9% might not be surprising. What other variables did you account for in your adjusted analysis? Please include them. You can say “After accounting for ……………, long duration on ART was significantly associated with presence of HIVDR mutations (aOR=; 1.15 95%CI 1.05 -1.26 p=0.003).”

HIVDR is conducted routinely for all PLHIV failing on ART effective January 2023. The other variables accounted for, have been indicated as ‘’ gender, age, NRTI anchor drug, ART regimen type and WHO clinical stage of the participants’’ (page 2 line 12-14)

Conclusion: replace “region” with a specific place being referred to. In the conclusion, there is no need of comparing with other previous findings. In this section, generally, you have to highlight your main findings and, the implication/s of those findings and end by giving recommendation/s. please revise this section accordingly.

The conclusion section in the abstract has been re-written to highlight main findings of high HIVDR, high etravirine and DTG resistance, low DRV resistance and recommendations. ( page 2, line 19-22)

Introduction

- In this section, I would first introduce the topic to the readers; in this case, HIVDR, give global and regional statistics.

The introduction has been re-written, starting with introducing HIVDR as guided.( page 3 line 5-11)

- Replace amongst with among

The word ‘amongst’ has been replaced with ‘among’, across the whole manuscript as guided. (Page 11 line 12; 12 line 4: 17 line 8: 18 line 12 and 15 )

Materials and Methods

- Say “retrospective cross-sectional…..”

This has been indicated as guided (page 5 line 3)

- Include the age of the participants, e.g., PLHIV aged � 15 years, etc.

The age of participants has been included. All participants irrespective of age were considered. (page 6 line 6)

- Define PCR

The PCR definition has been included as ‘’a molecular technique used to amplify specific DNA sequences by generating millions of copies from a small initial sample and is crucial for increasing the quantity of genetic material required for sequencing’’. (page 5 line 15-17)

- Data collection: How did you pre-test the data abstraction tool? Say data were (see my comment above)

We have Indicated that the data abstraction tool was pretested on 30 participants’ records. Replaced ‘was’ with ‘were’. (page 6 line 18-19)

- Ethics approval and consent to participate should be the last part of your materials and methods section. Please revise.

This ethics section has been moved to be the last part of the methods section as guided. (Page 7 line 15-23)

Data Analysis: state the test of normality used. Delete this part “The dependent variable was presence of at least one HIVDR mutation” and have a section in the materials and methods section where you will have “Operation definition” for your dependent/outcome/response variable.

Indicated that the Shapiro wilk test was used to assess normality of continuous data. (page 7 line 6-9)

Operational definitions have been included in the methods section as guided (Page 5 line 2-16).

Results

- “They had been on ART for a median 8 years”, who?

This has been changed to “ the participants had been on ART for a median of 8 years.’’ (page 11 line 2)

- Define in the methods section, preferably under “Operational definition” how age was categorized and cite.

A sub-title for Operational definitions has been created under the methods and material section. age was classified as follows: Children (<10 years), Adolescents (10–19 years) and Adults (≥20 years) according to WHO classification [Organization WH. Consolidated guidelines on HIV prevention, testing, treatment, service delivery and monitoring: recommendations for a public health approach: World Health Organization; 2021.] (Page 5 line 2-16).

- Table 1, I would name it “Baseline Participant characteristics and HIVDR”. Then, narrate the results of HIVDR also so you can guide the readers.

Table 1 has been renamed as guided and HIVDR results narrated (page 9 line 1) .

On variable ‘HIVDR Indication”, why didn’t you aggregate 2nd and 3rd lines based on children, adolescents, and adults just the way you did for 1st line? In the footnote of the

Thank you for the observation. Unfortunately, we used secondary data from the National database where the indication for HIVDR was not disaggregated according to age for those who were failing on 2nd and 3rd line.

Table, please define all the abbreviations.

Abbreviations have been included in the footnotes for the 4 tables as guided (pages 10, 11, 12 and 15).

- For Table 1, authors should conduct a chi-square test/Fisher’s exact test between HIVDR (yes or no) and other independent categorical variables and a Wilcoxon rank-sum test for two medians. Create 5 columns: column 1 Characteristics, column 2 Frequency, column 3 Presence of HIVDR, column 4 Absence of HIVDR, and column 5 p-value. We can now know whether the proportions and medians are statistically significant.

Table 1 has been redesigned as guided. We used chi square test to obtain P-Values for categorical variables and Wilcoxon rank sum test to obtain P-values for continuous variables which are not normally distributed. The same factors initially found significant at bivariate logistic regression were also significant using the above tests and were considered for multivariate analysis. (Page 9-10)

- Presence of HIVDR mutations: define CI. Show these results in the Table “HIVDR mutations were observed across different drug classes in the same individuals where 55% (161) of all participants had resistance to at least two drug classes; 20% (59) had to at least three drug classes and 1% (03) of the participants had resistance across all the 4 drug classes. The commonest cross resistance was observed between NNRTI and NRTI drug classes (67%).”

CI has been defined as the range within which the true population proportion is expected to fall with a 95% certainty (page 11Line 6-8).

Table 3 which shows the level of cross resistance across different drug classes in the same individuals has been included in the manuscript (Page 12).

- HIVDR Mutations amongst participants: Replace the word “amongst with among” throughout the document.

The word ‘amongst’ has been replaced with ‘among’, across the whole manuscript as guided.

- I don’t think Figures 1 and 2 should be supplementary. They have to be in the main document.

Fig 1 and Fig 2 have been included in the main manuscript document (Page 7 and 12).

- Factors associated with HIVDR: replace the word spends with spent.

The word “spends” has been replaced with “spent” as guided. (page 13 line 14).

In the abstract, you have to mention all the variables which were significant at adjusted analysis.

We have included the 2 factors of: indication for HIVDR test and Year of sample collection in the abstract results section as guided (Page 1 line15-17 ).

For 3rd line, the CI is too wide; I hope your discussion accounted for this imprecision.

After re-running the analysis following recategorization of age groups to align with WHO age classification, 3rd line was not significantly associated with HIVDR (Table 4 page 15).

Make sure all the p-values have three decimal places in Table 3. The footnote of all your Tables should define the abbreviations.

P-values in table 1 and 3 have been adjusted to three decimal places as guided (Page 10-15).

Discussion

- AZT including, say “included”

Addressed as guided, (page 17 line 14)

- noted by the program- say HIV program

Addressed as guided, (page 18 line 1)

- This does not read well “For the indication for the HIVDR test”, replace. Maybe “In case of the HIVDR test indication”

Addressed as guided, ( page 18 line 10)

- development HIVDR, here “of” is missing.

“Of” has been indicated as guided, (page 18 line 12)

- Delete respectively “prior regimen changes due to probable or confirmed HIVDR respectively”

Respectively has been deleted as guided. (page 18 line 14)

- Being a retrospective study, did you collect all the factors associated with HIV drug resistance e.g. CD4? You have only mentioned one limitation. Please include all the limitations inherent in retrospective studies.

The limitation section has been improved to indicate factors not assessed due to use of secondary data and limitations due to retrospective studies. (Page 19 line 9-14)

Conclusion

- Don’t think 73.9% is lower. Here, tell us the meaning (low or high) of the proportions you have found without comparing with previous studies.

The conclusion has been re-written to highlight major findings of high HIVDR, high etravirine and DTG resistance, low DRV resistance, implications of reduced treatment options and recommendations. (Page 19 line 16-22)

Reviewer #2: The manuscript had described a relatively lower prevalence of HIVDR among PLHIV with non-suppressive ART than other studies. It's notable that longer duration of ART was associated with presence of HIVDR mutations. However, this study had some shortcomings and should be reversed properly, then later considered if it is suit to published in this journal

Major concerns:

The topic should be clearer to define the study population that the prevalence of HIVDR was not among all PLHIV but among those with non-suppressive ART.

The title has been adjusted to show that the study applies to only those with virological failure.(Page 1 line 1)

The study had included children and adolescent which might have different prevalence of virological failure, HIVDR and related factors, and deserve to be described and discussed disparately.

Thank you for your comment. Indeed, children and adolescents may have different prevalence rates of virological failure, HIV drug resistance (HIVDR), and associated factors, which merit separate consideration and discussion.

We have included a subsection under results section titled HIVDR among children and adolescents where findings among Children and Adolescents were separately described and compared to the overall study findings. From literature review, children have a lower VL suppression rate in Uganda (63%) compared to the national average (79%) however from our logistic regression analysis, age was not independently associated with HIVDR. Full details have been indicated in the manuscript in the following sections: Introduction (page 3, line 21-22); Results section (page 15 and 16 ) and discussion section,( page 18, line 21)

Minor concerns:

There was not any rates of virological failure among different groups, which would also be related to non-adherence.

The study did not ascertain the VL failure rates since it considered only those with virological failure. However, we have indicated the different failure rates by age and gender in the introduction where adult males and children had a higher VL failure rate compared to national average (Page 3, line 21)

What was the prevalence of HIVDR among ART naive PLHIV in the regions.

HIVDR testing for ART naïve PLHIV is not routinely done in Uganda and hence no data was available at the national HIVDR database. We have however Indicated some literature on TDR in the introduction section. The Transmitted drug resistance rates were highest among NNRTI with global rates being at 12.9% and that of Uganda being higher at 15.4%. Other drug classes are fairly spared (Introduction, page

---

## [Decision Letter · Decision Letter 1]

28 May 2025

Dear Dr. Francis,

Thank you for submitting your manuscript to PLOS ONE. After careful consideration, we feel that it has merit but does not fully meet PLOS ONE’s publication criteria as it currently stands. Therefore, we invite you to submit a revised version of the manuscript that addresses the points raised during the review process.

We look forward to receiving your revised manuscript.

Kind regards,

Felix Bongomin, MB ChB, MSc, MMed, FECMM

Academic Editor

PLOS ONE

Journal Requirements:

Reviewers' comments:

Reviewer's Responses to Questions

**Comments to the Author**

Reviewer #1: All comments have been addressed

2. Is the manuscript technically sound, and do the data support the conclusions?

Reviewer #1: Yes

3. Has the statistical analysis been performed appropriately and rigorously?

Reviewer #1: Yes

4. Have the authors made all data underlying the findings in their manuscript fully available?

Reviewer #1: (No Response)

5. Is the manuscript presented in an intelligible fashion and written in standard English?

Reviewer #1: No

Reviewer #1: Introduction

- “ranging from 5.9% and 18.2%”, replace “and” with “to”.

Results

- Delete “Figure 1: Enrollment study profile”, since the figure has already been named below the figure.

- Narrations on the results should be above the tables

- Here “median age of 19 years” include the interquartile range (IQR)

- No need of this detail “where CI represents the range within which the true population proportion is expected to fall with a 95% Certainty”, please delete.

- Separate the narrations for each Table. Please make sure that each narration is above the table being referred to. Do the same for all figures. Follow what you did for Table 4.

- HIVDR among children and Adolescents: this part is confusing, please narrate your results above the Tables and Figures.

- Please restructure your results section.

**Do you want your identity to be public for this peer review?** For information about this choice, including consent withdrawal, please see our Privacy Policy

Reviewer #1: No

---

## [Author Response · Author response to Decision Letter 2]

21 Jun 2025

Review Comments to the Author

Reviewer #1: Introduction

- “ranging from 5.9% and 18.2%”, replace “and” with “to”.

We have replaced ‘and’ with ‘to’ as guided (see page 4 line 12)

Results

- Delete “Figure 1: Enrollment study profile”, since the figure has already been named below the figure.

The title, Figure 1: Enrollment study profile has been deleted as guided (Page 8 line 4)

- Narrations on the results should be above the tables

The narrations for all tables have been put above the tables as guided (See page 9 - 12)

- Here “median age of 19 years” include the interquartile range (IQR)

The interquartile range (IQR) of 13–41 years has been indicated (Page 9 line 1-2)

- No need of this detail “where CI represents the range within which the true population proportion is expected to fall with a 95% Certainty”, please delete.

The above statement has been deleted as guided (Page 9 line 7)

- Separate the narrations for each Table. Please make sure that each narration is above the table being referred to. Do the same for all figures. Follow what you did for Table 4.

The narrations for all tables have been put above the tables as guided (Page 9 and 12)

- HIVDR among children and Adolescents: this part is confusing, please narrate your results above the Tables and Figures.

We have included the narrative of children and adolescents above table 4 as guided (Page 5).

- Please restructure your results section.

Results have been restructured as guided above

---

## [Editor Report · Decision Letter 2]

25 Jun 2025

Prevalence of HIV drug resistance, its correlates and common mutations among people living with HIV failing on ART in northern Uganda. A cross-sectional study.

PONE-D-25-09503R2

Dear Dr. Francis,

We’re pleased to inform you that your manuscript has been judged scientifically suitable for publication and will be formally accepted for publication once it meets all outstanding technical requirements.

Kind regards,

Felix Bongomin, MB ChB, MSc, MMed, FECMM

Academic Editor

PLOS ONE
---

## [Editor Report · Acceptance letter]

PONE-D-25-09503R2

PLOS ONE

Dear Dr. Francis,

I'm pleased to inform you that your manuscript has been deemed suitable for publication in PLOS ONE. Congratulations! Your manuscript is now being handed over to our production team.

Kind regards,

on behalf of

Dr. Felix Bongomin

Academic Editor

PLOS ONE